# SPIN modification for low temperature experiments

André Welti[1], Kimmo Korhonen[2], Pasi Miettinen[2], Ana A. Piedehierro[1], Yrjö Viisanen[1], Annele Virtanen[2], and Ari Laaksonen[1,2]

[1]Finnish Meteorological Institute, Helsinki, Finland
[2]Department of Applied Physics, University of Eastern Finland, P.O. Box 1627, 70211, Kuopio, Finland

*Correspondence to:* A. Welti (andre.welti@fmi.fi)

**Abstract.** The SPectrometer for Ice Nuclei (SPIN) has been modified to access ice nucleation at low temperatures. The modification consists of a reconfiguration of components from SPIN's cooling system to provide refrigerant with a low boiling point to the chamber. We describe the setup modification and determine the temperature and humidity range accessible to experiments. The modification extends the measurement range of SPIN to 208 K, which enables measurements in the temperature regime relevant for ice formation in cirrus clouds. This addition of low temperature capability allows for far more comprehensive measurements of the temperature- and humidity- dependent ice nucleation of test substances, to investigate fundamentals of ice nucleation mechanisms. We present exemplary data of heterogeneous ice nucleation on silver iodide and homogeneous ice nucleation in solution droplets to demonstrate the usefulness of the modified SPIN setup for precision measurements to detect discrepancies between experiments and widely used theories.

## 1 Introduction

The fundamental understanding of atmospheric ice formation is a complex problem that has been under investigation for almost a century (Findeisen, 1938). Part of the complexity of studying ice nucleation experimentally arises from different mechanisms that initiate ice formation at different temperatures and humidities. Tropospheric ice nucleation at low temperatures ($T < 236\,\mathrm{K}$), typical for cirrus clouds, proceeds at water sub-saturated conditions by homogeneous nucleation of aqueous aerosol or heterogeneous nucleation from the vapour phase. At intermediate temperatures ($236\,\mathrm{K} < T < 273\,\mathrm{K}$) heterogeneous ice nucleation above water saturation leads to glaciation of mixed-phase clouds (Pruppacher and Klett, 1997). Experimental work on heterogeneous ice nucleation mechanisms include the study of a variety of substances and their ice nucleation potential (e.g. Hoose and Möhler, 2012), both at water saturated (immersion freezing, contact freezing) and sub-saturated conditions (deposition ice nucleation, (pore-) condensation freezing). Experimental techniques to study each ice nucleation mechanism have been developed over the years (e.g. DeMott et al., 2011). The continuous flow diffusion chamber (CFDC) type of experiment has proven to be a versatile method to isolate and observe different mechanisms, especially ice nucleation from the vapour phase. As Rogers (1988) pointed out, the advantage of CFDCs is the separate control over temperature and humidity in contrast to expansion or mixing chambers for which T and relative humidity (RH) are interdependent. Often, the dependency of ice nucleation on T and RH by a specific mechanism, can be predicted from theoretical considerations (e.g. Fletcher, 1962; Koop et al., 2000; Marcolli, 2020). Experimental characterization of T-, RH-dependent ice formation is a tool to validate and

refine the theories on ice nucleation mechanisms.

We describe a mechanically uncomplicated way to modify the cooling system of the SPIN instrument to expand its T and RH range in which experiments can be performed. The modification is beneficial for laboratory studies investigating ice nucleation in a broad T and RH range.

## 5  2   The SPectrometer for Ice Nuclei (SPIN)

SPIN is a parallel plate CFDC manufactured by Droplet Measurement Technologies (DMT). It follows the working principle discussed in Rogers (1988) with the parallel plate design from Stetzer et al. (2008). For a description of the original SPIN instrument and reference experiments we refer to Garimella et al. (2016). Here, the focus is on the compressor cooling system of SPIN, which is the component that was modified for low temperature operation. The cooling system generates high pressure,

liquid coolant used to cool and maintain temperature of the chamber wall plates by dosed evaporation of the coolant. In the following, we describe how experimental conditions (T, RH) are generated by controlling the wall plate temperatures, and how much the limits of achievable experimental conditions are extended by modifying the cooling system.

### 2.1   Operating principles

Experimental conditions (T, RH) in SPIN are created by coating the parallel (10 mm apart) wall plates of the chamber with

a thin (1 mm) ice layer acting as water reservoir, and individually controlling the temperatures of the front (warm) and back (cold) plate. The ice coated walls keep the vapour pressure boundary conditions at ice saturation with respect to the set wall temperature. Under steady state conditions a linear temperature and water vapour partial pressure gradient establishes between the plates. For an explicit derivation of the linear temperature and water vapour partial pressure field in a CFDC we refer to Rogers (1988), and Lüönd (2009). Because of the non-linear temperature dependance of saturation on water vapour partial

pressure (Clausius-Clapeyron), super-saturated conditions can be generated in the 8 mm gap between the ice covered walls by setting different wall temperatures. For experiments, a lamellar sample, which is confined by a sheath flow to a narrow position between the ice covered wall plates, is passed through the chamber. T and RH at the lamina position are controlled by the linear temperature and water vapour partial pressure gradient between the cold and warm plate. Fig. 1 shows exemplary temperature and water vapour partial pressure gradients in SPIN, at water saturation ($RH_w = 100\%$) and T=203 K, 213 K and 223 K at the

position of the sample lamina. A ready to use code to determine the position of the sample lamina can be found in Kulkarni and Kok (2012). For a discussion on sampling bias due to particle displacement outside the lamina we refer to Garimella et al. (2017), and Korhonen et al. (2020).

### 2.2   Modified cooling system

To control the chamber wall temperatures in the original SPIN setup, two independent refrigeration compressor cycles are used.

The warm wall is cooled by a single stage compressor cycle using refrigerant R404A, while the cold wall is cooled by a two stage (cascade) compressor system with refrigerant R404A in the first, higher stage and refrigerant R116 for the second, lower

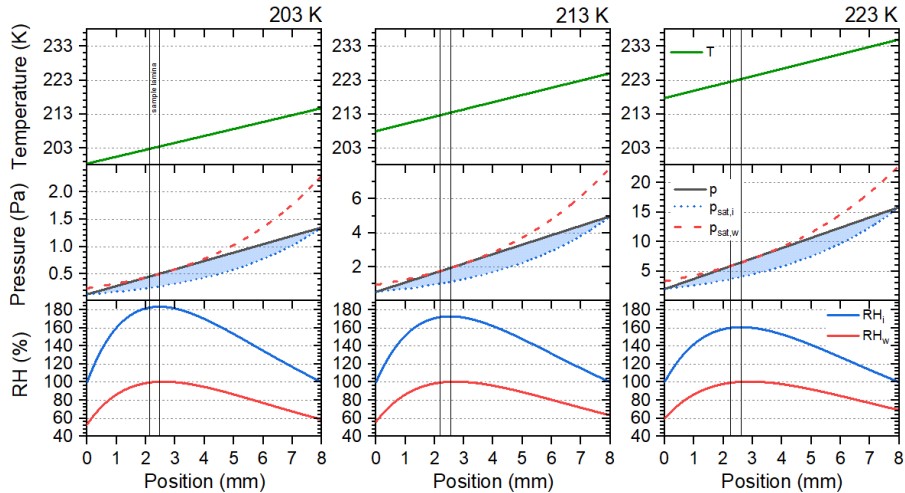

**Figure 1.** Linear temperature (top row) and water vapour partial pressure (second row) gradients between the wall plates of SPIN at 203 K, 213 K, 223 K lamina temperature and $RH_w = 100\%$. Water vapour partial pressure ($p$) is compared to the saturation vapour pressure over water ($p_{sat,w}$) and ice ($p_{sat,i}$), calculated according to Murphy and Koop (2005). Ice super-saturated conditions are indicated as shaded areas. The resulting profiles of relative humidity with respect to ice ($RH_i$) and with respect to water ($RH_w$) are given in the third row. Sample position is indicated by vertical lines.

stage (see Fig. 2 for a cascade compressor diagram). Refrigerant R116 has a boiling point of 194.95 K (at 1 atm pressure), refrigerant R404A has a boiling point of 226.65 K (at 1 atm pressure). The single stage compressor cycle using R404A to cool the warm wall plate in the original setup limits the range of T and RH achievable at the sample lamina position. To reach lower temperatures, the SPIN cooling system has been modified by reconnecting the cold wall, cascade compressor system to deliver R116 refrigerant to both wall plates. The configuration of the modified setup is shown in Fig. 2. In practice, three modifications to the cooling system are needed:

1. At the upper part of the chamber, a junction is added to the high pressure liquid R116 line (lower stage, yellow line in Fig. 2) to connect it to the warm wall plate in parallel to the cold wall.

2. The refrigerant outlet lines of cold and warm wall, where the refrigerant exits the wall plates in the form of low pressure gas (lower stage, blue line in Fig. 2) are joint together to return to the cold 2 compressor.

3. Accounting for the increased volume of R116 needed to cool both wall plates, an additional expansion volume (14ℓ steel tank) is added to the return line (lower stage, blue line in Fig. 2), to give room to the gaseous refrigerant and prevent overpressure when the system is not running.

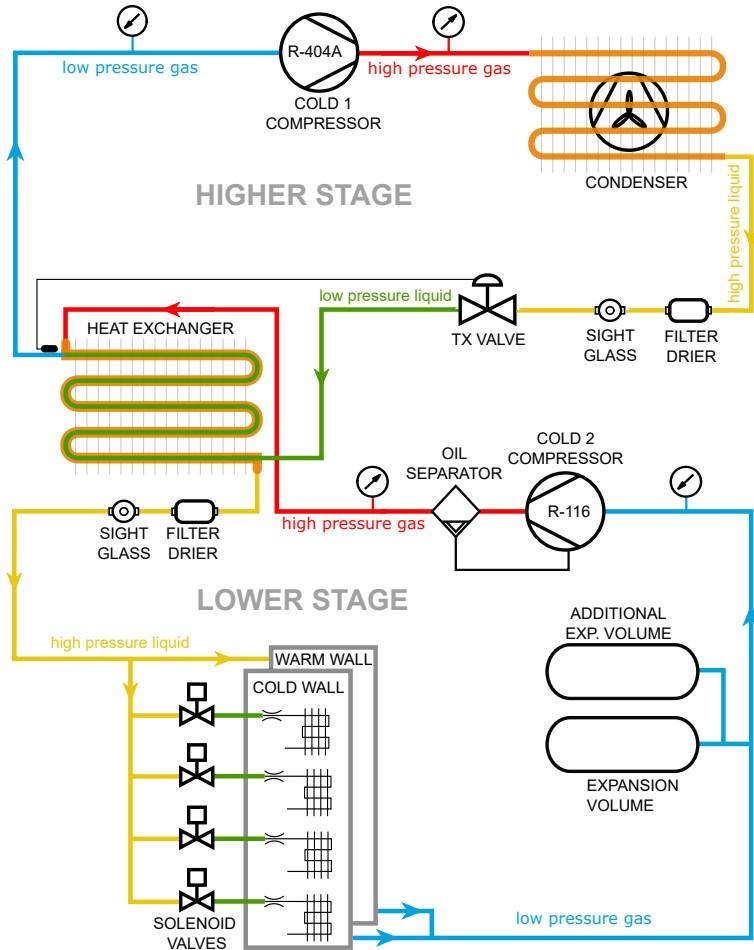

**Figure 2.** Diagram of two stage cascade compressor setup. Direction of refrigerant circulation is indicated by arrows. Pressure and phase state of the refrigerant is indicated by line colors. For the low temperature SPIN modification, R116 from the lower stage is used to cool both wall plates, and an additional expansion volume is added.

A consequence of using only one instead of two compressors to deliver the refrigerant for both walls, is a reduction in the achievable cooling rate from approximately $2\,\mathrm{Kmin^{-1}}$ to $1\,\mathrm{Kmin^{-1}}$ above $233\,\mathrm{K}$ and decreasing to $< 0.5\,\mathrm{Kmin^{-1}}$ towards the lowest temperatures. Simultaneous cooling of both walls is needed during cooling of the chamber to start an experiment or measurements in which temperature is changed at a constant humidity (T scan). For measurements in which humidity is varied at a constant lamina temperature (RH scan) only one wall plate is cooled while the other is heated. During such experiments, where most of the refrigerant is used to cool one wall, the cooling rate of original and modified setup are identical.

The range of experimental conditions achievable with the original and the modified cooling system of SPIN are shown in Fig. 3. Fig. 3(a) shows the theoretical limits of T and RH conditions, determined according to the examples shown in Fig. 1 by

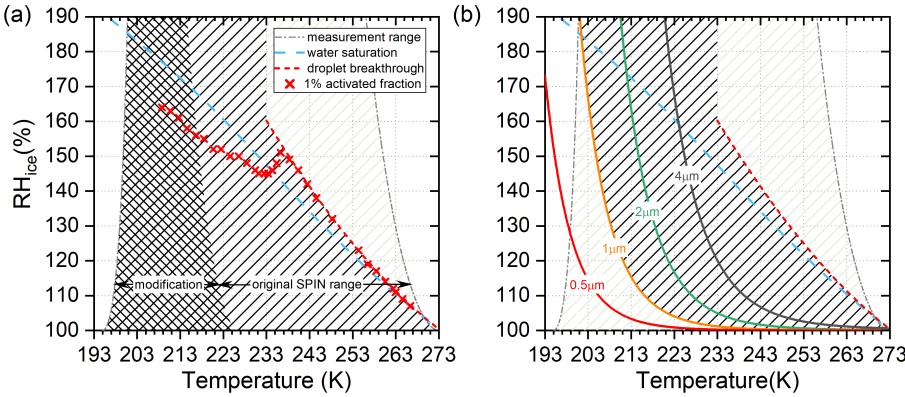

**Figure 3.** Calculated range of T-,RH$_i$-conditions. (a) Extended limit of experimental capacity with the modified SPIN cooling system (cross-hatched area) compared to the original setup (hatched area). Water saturation is indicated as blue dashed line. Red crosses show 1% activated fractions, measured with 200 nm ammonium sulfate particles (see Fig. 4). Above 235 K the data indicate onset of water breakthrough conditions, below 235 K homogeneous freezing of solution droplets. (b) Limiting conditions to distinguish 0.5 μm, 1 μm, 2 μm and 4 μm (diameter) test particles from growth limited ice crystals. Conditions excluded from ice detection by droplet breakthrough and ice crystal growth are lightly hatched. Total experimental range to measure ice nucleation with the modified SPIN setup is shown as dark hatched area.

varying the cold and warm wall temperatures from 273.15 K-194.95 K (ice melting temperature to boiling point of R116). The range of the original SPIN setup is calculated with the cold wall varying from 273.15 K-194.95 K and the warm wall between 273.15 K-226.65 K (boiling point of R404A). Note that ambient heat loss is not considered for the shown range. In practice, the achievable wall temperatures lie 5-10 K above the boiling point of the refrigerant (at 293 K ambient temperature). In addition to

instrumental limitations, the conditions under which experiments can be conducted are limited by the optical detection method used to distinguish ice crystals from test aerosol and droplets by size. The SPIN chamber comprises an isothermal section at the end in which conditions are maintained at ice saturation to evaporate droplets (Garimella et al., 2016). Fig. 3(a) shows the experimentally determined droplet breakthrough conditions where droplets growing above water saturation do not shrink below the size range where particles are classified as ice crystals. Droplet breakthrough limits the explorable humidity range of

ice nucleation down to temperatures at which droplets freeze homogeneously. A second limit to detect ice nucleation is caused by the ice crystal growth velocity, which slows down considerably towards lower temperatures due to the lower water vapour partial pressure (cf. second row of Fig. 1) at a certain relative humidity. Conditions for ice crystals to grow to a diameter of 0.5 μm, 1 μm, 2 μm and 4 μm within ∼ 10 s residence time in the ice super-saturated section of SPIN are calculated according to Rogers and Yau (1989) and shown in Fig. 3(b). Details on the calculation of the ice crystal size are included in Appendix A.

Ice nucleation experiments often use test aerosol with diameter up to 1 μm, thus limiting experiments to conditions where the test aerosol can be distinguished from ice crystals to above the 1 μm ice crystal growth conditions shown in Fig. 3(b). Note that droplet breakthrough line and ice crystal growth, limit the experimental conditions to the same T and RH in any CFDC with dimensions and flows similar to SPIN.

## 3 Laboratory performance

The modified SPIN setup is used to measure the ice nucleation activity of two reference materials in a T and RH range covering all tropospheric ice nucleation conditions. Experiments with ammonium sulfate $((NH_4)_2SO_4)$ are conducted to measure homogeneous ice nucleation and to determine the droplet breakthrough RH of the setup. Experiments with silver iodide $(AgI)$ are conducted as an example for the setups utility to investigate heterogeneous ice nucleation.

Aqueous $(NH_4)_2SO_4$ aerosol is generated with an atomizer (Aerosol Generator Model 3076, TSI) and subsequently dried below the efflorescence point, while AgI particles are dry generated by agitation of powder with a magnetic stirrer. For both test substances, experiments are performed using dry 200 nm aerosol particles, size selected with a differential mobility analyzer (DMA; TSI Model 3081).

### 3.1 Homogeneous freezing of ammonium sulfate solution

$(NH_4)_2SO_4$ is a prevalent aerosol throughout the troposphere and commonly used for instrument calibration. Although there are reports of $(NH_4)_2SO_4$ forming ice heterogeneously at cirrus temperatures (Abbatt et al., 2006), it is often applied as a hygroscopic test substance for forming aqueous aerosol to investigate homogeneous ice nucleation of solutes (e.g. Koop et al., 2000).

Fig. 4(a) shows the measured activated fraction (AF) of particles, calculated as the concentration ratio of detected ice crystals exiting SPIN to injected dry particles. Within the limits of detection of the SPIN experiment (approximately $AF \geq 1E-4$), no heterogeneous ice nucleation on solid $(NH_4)_2SO_4$ particles was observed. The very steep onset of ice formation in dependance of relative humidity below 235 K indicates homogeneous ice nucleation. A comparison of 1% AF conditions to literature data and 1% AF curve derived using the parametrization of Koop et al. (2000) are shown in Fig. 4(b). Details on the calculation of the homogeneous freezing line are given in Appendix B. While the data are in general agreement to previous observations, there is a clear offset between the slope of the Koop-line for 1% AF and the SPIN data. Homogeneous freezing is observed to set in at lower relative humidities just below 235 K and the temperature dependent increase in ice super-saturation necessary for constant activity is steeper (almost parallel to the water saturation line). Possible reasons for the discrepancy between the measurements in this study and the Koop-parametrisation are time dependent effects, i.e. aqueous aerosol do not reach equilibrium before freezing in SPIN.

### 3.2 Heterogenous ice nucleation on silver iodide

Silver iodide $(AgI)$ is an exceptionally active ice nucleating substance (Vonnegut, 1947). $AgI$ is widely used for cloud seeding operations to induce glaciation in supercooled liquid clouds, and also for seeding of cloud free, ice super-saturated parts of the atmosphere to generate cirrus clouds (Vonnegut and Maynard, 1952). $AgI$ has also served as a model substance to investigate the size and time dependance of ice nucleation and to develop the classical nucleation theory (CNT) of heterogeneous ice formation (Fletcher, 1959).

Measurements in the entire T-RH space are shown in Fig. 5(a). Fig. 5(b) shows a comparison of 1% AF conditions to literature

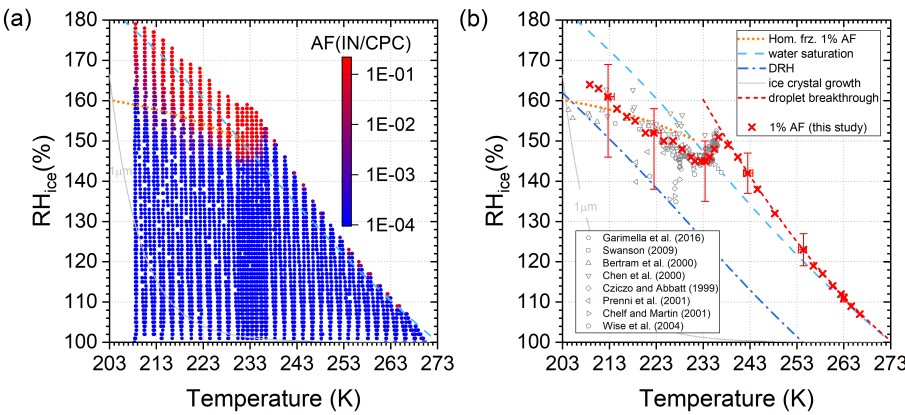

**Figure 4.** (a) Activated fraction of $200\,\mathrm{nm}$ $(NH_4)_2SO_4$ particles, (b) 1% AF in comparison to literature data (references indicated in the figure legend) and the parametrization given in Koop et al. (2000). Lines indicate the Koop-line for homogeneous ice formation, water saturation, deliquescence relative humidity (DRH, Seinfeld and Pandis, 2006), droplet breakthrough and 1 μm ice crystal growth conditions. The range of experimental variability in $\mathrm{RH}_{ice}$ and T is shown at $10\,\mathrm{K}$ intervals.

data. Most previous laboratory experiments were limited to T>233 K by the measurement devices. The data collected here show good agreement to the large amount of data in this temperature range. Below 233 K only Bailey and Hallett (2002) and Detwiler and Vonnegut (1981) previously reported data. The two studies used static diffusion chambers and reported ice nucleation onset conditions without specifying the activated fraction and conditions for 1% AF, respectively. While the two previous studies observed an inflection in the slope of ice super-saturation needed for a constant amount of ice formation with decreasing temperature, the current study shows a more monotonic increase in ice super-saturation to activate 1% of the monodisperse $AgI$ particles. A monotonic but less steep increase in ice super-saturation with decreasing temperature for a constant AF is predicted by CNT (Detwiler and Vonnegut, 1981).

## 4 Discussion

The modification of the cooling system of SPIN increases the experimentally accessible conditions to the entire T and RH range of ice and mixed-phase cloud formation in the troposphere. Overall, the modification maximizes the measurement range of a CFDC type instrument with the dimensions of SPIN, since the growth of ice crystals limits the lowest temperature at which ice crystals can be detected and distinguished from dry aerosol (shown in Fig. 3(b)). In comparison to the original SPIN setup, it allows a broader study of ice nucleation at lower temperature conditions, where ice super-saturation extensively occurs in the atmosphere and ice crystals form before humidities approach water saturation (Detwiler and Vonnegut, 1981).

To evaluate the performance of the modified SPIN for measuring homogeneous and heterogeneous ice nucleation, experiments using size selected, $200\,\mathrm{nm}$ $(NH_4)_2SO_4$ and $AgI$ particles are compared to extensive literature data (see Fig. 4 and Fig. 5).

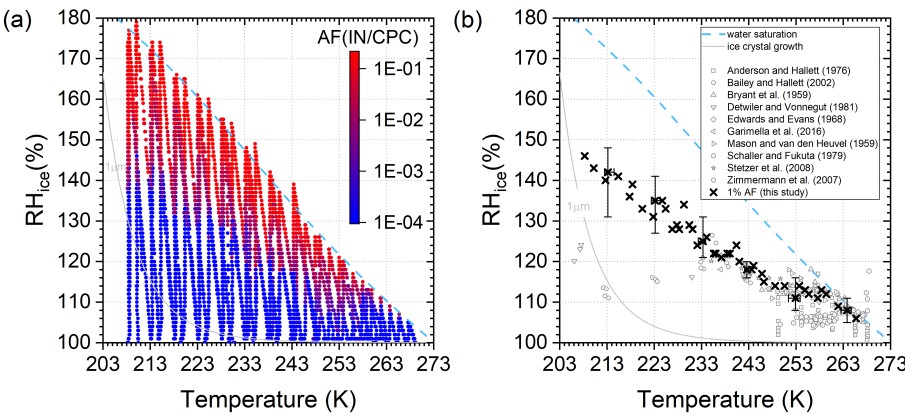

**Figure 5.** (a) Activated fraction of $200\,\mathrm{nm}$ $AgI$ particles, (b) 1% AF in comparison to literature data (references indicated in the figure legend). Water saturation and $1\,\mu\mathrm{m}$ ice crystal growth conditions are indicated as lines. The range of experimental variability in $\mathrm{RH}_{ice}$ and T is shown at $10\,\mathrm{K}$ intervals.

While the data are generally consistent, the measurements of this study show systematic discrepancies in T-, RH-dependance both from the predicted heterogeneous ice nucleation of CNT (Fletcher, 1962) and from the widely used parametrisation of homogeneous ice nucleation of solution droplets by Koop et al. (2000). Apart from differences in the method of particle gener-
ation, size segregation and detection, it is unclear why partly systematically deviating T-, RH-dependencies were observed in similar experiments. It highlights the need for comprehensive ice nucleation data sets, measured for a wide range of substances, to improve the understanding of natural and artificial ice nucleation. The automation of the SPIN setup in combination with the extended measurement capability through the modification described here can be a tool to lay the experimental groundwork.

## 5   Conclusions

We describe a mechanically simple modification of the cooling system of the commercial SPIN instrument. Compared to
the cooling system of the original SPIN setup, the modified system has a reduced cooling rate during initial cooling of the chamber in preparation for an experiment, however cooling rates during experiments are identical. In practice, the modification extends the experimental possibilities by more than $20\,\mathrm{K}$ to the full T and RH range of ice nucleation under tropospheric cirrus cloud conditions. This additional temperature range is particularly useful for experiments investigating the temperature- and humidity- dependence of ice nucleation, since a number of aerosol species only efficiently trigger ice nucleation at such low
temperatures. The usefulness of the modified SPIN instrument for laboratory studies, is exemplified by characterising the T-, RH-dependence of heterogeneous ice nucleation on $AgI$ particles and homogeneous ice nucleation in $(NH_4)_2SO_4$ solution droplets. From both new data sets a systematic discrepancy between experimentally measured and theoretical predicted ice nucleation is discovered. With the modified SPIN instrument, specific ice nucleation mechanisms can be studied in a broader

T and RH range. The instrument is useful to validate theoretical aspects of ice nucleation and it can be used to investigate ice nucleation on natural and artificial aerosols.

*Data availability.* Data sets are available from the authors upon request.

*Author contributions.* AW, KK and PM modified the chamber. AW conducted the experiments with contributions from KK and AAP. AW
5   prepared the manuscript with contributions from KK, AAP and AL. All authors commented the manuscript. YV, AV and AL acquired funding and supervised the project.

*Competing interests.* The authors declare that they have no conflict of interest.

*Acknowledgements.* This work was supported by the Academy of Finland, C-Main project (grant no. 309141), and the Centre of Excellence program (grant no. 30733). We thank Angela Buchholz, Scott Ohde and Frank Sagan for helpful discussions. We acknowledge technical
10   assistance from Mikko Hartikainen.
Edited by:

## Appendix A: Ice crystal growth

The growth of ice crystals is estimated from the mass growth rate, $\frac{dm}{dt}$ (Rogers and Yau, 1989; Lohmann et al., 2016):

$$\frac{dm}{dt} = \alpha \cdot 4 \cdot \pi \cdot C \cdot \left( \frac{S_i - 1}{F_k + F_d} \right), \tag{A1}$$

where $\alpha$ is the accommodation coefficient for water molecules (Skrotzki et al., 2013), $S_i$ the saturation ratio with respect to
ice, and $C$ the particle capacitance which incorporates the size and shape of the ice crystal (Lohmann et al., 2016). Assuming
a spherical shape of the ice crystal for simplicity, $C$ equals its radius, $C = r$ (see Lamb and Verlinde, 2011, for $C$ for different
shapes). $F_k$ and $F_d$ in the denominator of Eq. A1 are

$$F_k = \left( \frac{L_s}{R_v \cdot T} - 1 \right) \cdot \frac{L_s}{K \cdot T}, \tag{A2}$$

$$F_d = \frac{R_v \cdot T}{D_v \cdot p_{sat,i}}, \tag{A3}$$

where $L_s$ is the latent heat of sublimation (Murphy and Koop, 2005), $R_v$ is the individual gas constant for water vapour
(Rogers and Yau, 1989), $T$ is temperature, $K$ is the thermal conductivity of air (Tsilingiris, 2008), $D_v$ the water vapour
diffusion coefficient in air (Hall and Pruppacher, 1976), and $p_{sat,i}$ is the saturation vapour pressure over ice (Murphy and
Koop, 2005).

Substituting $\frac{dm}{dt}$ with $\rho_i \cdot 4\pi r^2 \frac{dr}{dt}$ in Eq. A1 and rearranging leads to the growth equation in terms of the ice crystal radius:

$$r \frac{dr}{dt} = \alpha \cdot \left( \frac{S_i - 1}{\rho_i \cdot (F_k + F_d)} \right), \tag{A4}$$

where $r$ is the ice crystal radius, and $\rho_i$ is the mass density of ice.

Integration yields the time dependent radius of an ice crystal:

$$r = \sqrt{r_0^2 + 2 \cdot \alpha \cdot \left( \frac{S_i - 1}{\rho_i \cdot (F_k + F_d)} \right) \cdot t}, \tag{A5}$$

where $r_0$ is the seed particle radius, and $t$ is time.

## Appendix B: Homogeneous freezing of $(NH_4)_2SO_4$ solution droplets

The fraction of $(NH_4)_2SO_4$ solution droplets expected to freeze according to the water activity ($a_w$) dependent, homogeneous
nucleation rate parametrization by Koop et al. (2000) is given by

$$FF = 1 - exp(-J_{hom}(a_w) \cdot V_d \cdot t), \tag{B1}$$

where $FF$ is the frozen fraction, $J_{hom}(a_w)$ the nucleation rate, $V_d$ the volume of the solution drop, and $t$ is time. The volume $V_d$ of $(NH_4)_2SO_4$ solution droplets can be calculated based on the $a_w$ dependent growth factor ($GF$) reported in Wise et al. (2003):

$$GF = 1.49 + 2.81 \cdot (a_w)^{24.6}, \tag{B2}$$

leading to

$$V_d = \frac{4}{3} \cdot \pi \cdot (r \cdot GF)^3, \tag{B3}$$

with $r$ the radius of the dry $(NH_4)_2SO_4$ particle.

The parametrization of $J_{hom}(a_w)$ is reproduced from Tab. 1 in Koop et al. (2000):

$$\log_{10}(J_{hom}) = -906.7 + 8502 \cdot \Delta a_w - 26924 \cdot (\Delta a_w)^2 + 29180 \cdot (\Delta a_w)^3, \tag{B4}$$

with

$$\Delta a_w = a_w \cdot \exp\left(\frac{\int (v_w - v^i)dp}{R \cdot T}\right) - a_w^i, \tag{B5}$$

where $R$ is the ideal gas constant. The integral can be approximated by

$$\int (v_w - v^i)dp \approx v_w^0 \cdot (p - \frac{1}{2} \cdot \kappa^0 \cdot p^2 - \frac{1}{6} \cdot \frac{\partial \kappa^0}{\partial p} \cdot p^3) - v_i^0 \cdot (p - \frac{1}{2} \cdot \kappa^i \cdot p^2 - \frac{1}{6} \cdot \frac{\partial \kappa^i}{\partial p} \cdot p^3), \tag{B6}$$

with $v_w^0 = -230.76 - 0.1478 \cdot T + \frac{4099.2}{T} + 48.8341 \cdot \ln(T)$,     $v_i^0 = 19.43 - 2.2 \times 10^{-3} \cdot T + 1.08 \times 10^{-5} \cdot T^2$,

$\kappa^0 = 1.6 GPa^{-1}$, $\frac{\partial \kappa^0}{\partial p} = -8.8 GPa^{-2}$, $\kappa^i = 0.22 GPa^{-1}$, and $\frac{\partial \kappa^i}{\partial p} = -0.17 GPa^{-2}$, where $\kappa^0$, $\kappa^i$ are the isothermal compressibility of water and ice at ambient pressure. $v_w^0$ and $v_i^0$ are the molar volume of liquid water and hexagonal ice at ambient pressure.

The internal droplet pressure $p[GPa]$ in Eq. B6 can be calculated using the $GF$ from Eq. B2 and the surface tension of the solution droplet ($\sigma_{sol}$):

$$p = \left(\frac{2 \cdot \sigma_{sol}}{r \cdot GF} + p_{sat,w}\right) \times 10^{-9}, \tag{B7}$$

where $p_{sat,w}$ is the saturation vapour pressure over water (Murphy and Koop, 2005). $\sigma_{sol}$ is given in (Seinfeld and Pandis, 2006):

$$\sigma_{sol} = 0.0761 - 1.55 \times 10^{-4} \cdot (T - 273) + 2.17 \times 10^{-3} \cdot M, \tag{B8}$$

where $M$ is the molarity of $(NH_4)_2SO_4$ in the droplet which can be determined from the $GF$.

$a_w^i$ in Eq. B5 is given by

$$a_w^i = \exp\left(\frac{210368 + 131.438 \cdot T - \frac{3.32373 \times 10^6}{T} - 41729.1 \cdot \ln(T)}{R \cdot T}\right). \tag{B9}$$

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
