# Peer review of "SPIN modification for low temperature experiments"

_Atmospheric Measurement Techniques, 2020_

## Referee Comment (RC1) · Anonymous Referee #2 · 14 Aug 2020

**SPIN modification for low temperature experiments.**

André Welti, Kimmo Korhonen, Pasi Miettinen, Ana A. Piedehierro, Yrjö Viisanen, Annele Virtanen, and Ari Laaksonen

https:// doi.org/10.5194/amt-2020-215

**Summary**

This study introduces a new design that allows the Spectrometer for Ice Nuclei, SPIN, to investigate ice nucleation at colder temperatures. The demonstrated technique involves using one refrigerant loop to cool both walls of the chamber. The authors demonstrate that their simple design modification extends the temperature range by approximately 20 degrees below the range traditionally used by the SPIN community. The performance of the new design is evaluated using ammonium sulfate ($(NH_4)_2SO_4$) and silver iodide (AgI).

**General Comments**

The paper summarizes a design modification that will be of significant interest to other SPIN users and most likely other continuous flow diffusion chamber (CFDC) designs as well. Their technique has several demonstrable benefits. The authors note that the SPIN can now operate within the full set of conditions relevant to cirrus formation in the upper troposphere. One additional benefit not explicitly highlighted by the authors is the reduced number of compressors. Previous attempts to operate CFDCs at cold temperatures have resulted in the damage of compressors, and reducing the number of compressors used in the design will likely reduce instrument down-time due to repairs.

This paper is therefore well-suited for publication in AMT. Below, I provide a few questions and comments to strengthen the paper and clarify the interpretation of the SPIN results. The authors will note that most of these are minor points. I therefore recommend the paper be accepted for publications after the authors have adequately addressed or responded to them.

**Major Comments**

1. My only major comment concerns the interpretation of the apparent early-onset of homogeneous freezing of $(NH_4)_2SO_4$. The authors note that the $(NH_4)_2SO_4$ solution droplets appear to nucleate at a lamina RH lower than that expected for homogeneous freezing ((Koop et al., 2000)). The authors claim a discrepancy between their data and the Koop parameterization. However, I would venture a guess that their data actually presents no discrepancy if the uncertainty in lamina RH is taken into account.
   a. I believe Fig. 4 reports the average lamina conditions. However, at high RH and cold temperatures, CFDCs (and I would guess SPIN) generally show a few % uncertainty in the lamina RH. This uncertainty is caused by variability in the wall temperatures. Colder regions of the wall – for example, where refrigerant is injected – can cause certain areas of the lamina to experience a higher RH than the average. See e.g. Kulkarni and Kok, 2012, for a simple method and pre-written code to calculate the variability in lamina RH for a CFDC with SPIN's geometry.
   b. If this is lamina range is taken into account, does the onset of $(NH_4)_2SO_4$ nucleation more closely align with the Koop homogeneous freezing parameterization?

**Minor Comments**
**Abstract**

2. Page 1 Lines 4-5: "*The modification extends the measurement range of SPIN by more than 20 K to the temperature regime relevant for ice formation in cirrus clouds.*" Can the authors specify in the abstract the lower temperature range now achievable with their design modification?

**Introduction**

3. Page 1 Lines 13-15: "*Tropospheric ice nucleation at low temperatures (T < 236 K), typical for cirrus clouds, proceeds at water sub-saturated conditions by homogeneous nucleation of aqueous aerosol or heterogeneous nucleation from the vapour phase.*" The authors should briefly mention the possibility that heterogeneous nucleation below water supersaturation could be due to the pore-condensation freezing mechanism. E.g. (David et al., 2019; Marcolli, 2014)

**Operating Principles**

4. Page 2 Lines 17-18: "*For an explicit derivation of the linear temperature and vapour pressure field in a CFDC we refer to Rogers (1988); Lüönd (2009).*" I recommend also citing here Kulkarni and Kok, 2012 – it specifically discusses calcualtion of lamina conditions for the parallel plate (SPIN) design.

5. Page 2 Line 21: "*…a lamellar sample, which is confined by a sheath flow to a narrow position between the ice covered wall plates…*" The authors should briefly note here recent work that demonstrates aerosol samples are \*not\* constrained by sheath flows but rather spread outside the lamina. See e.g. DeMott et al., 2015; Garimella et al., 2017. This fact should not much change the author's results or interpretation, but it is important for the field of CFDC users to start to acknowledge.

**Modified Cooling System**

6. Page 3 Line 31: "*To reach lower temperatures, the SPIN cooling system has been modified by reconnecting the cold wall, cascade compressor system to deliver R116 refrigerant to both wall plates.*" The authors may know that R116 is a HFC whose use is being phased out in the European Union as per the Kigali Amendment to the Montreal Protocol. At this time, do the authors have knowledge of any non-HFC refrigerant (e.g. hydrofluoroolefin, $CO_2$) that might be an acceptable substitute in the future? The authors might note that if HFC refrigerants are banned, a new overall to the SPIN or other CFDC instruments' refrigeration loops may be needed anyways.

7. Page 3 Lines 4-6: "*A consequence of using only one instead of two compressors to deliver the refrigerant for both walls, is a reduction in the achievable cooling rate from approximately 2 K min⁻¹ to 1 K min⁻¹ above 233 K and decreasing to < 0.5 Kmin−1 towards the lowest temperatures.*" Please comment on the circumstances when both walls need to be cooled concurrently (i.e. during cooling down to start experiment).

8. Page 3 Lines 11-13: "*The range of the original SPIN setup is calculated with the cold wall varying from 273.15 K-194.95 K and the warm wall between 273.15 K-226.65 K (boiling point of R404A).*" The authors should note how this range compares to the coldest temperatures previously achieved for SPIN experiments. The coldest I can find published are ~-58 ˚C (Wolf et al., 2019) and ~-56 ˚C (Nichman et al., 2019).

9. Page 4 Lines 7-8: "*Conditions for ice crystals to grow to a diameter of 0.5 µm, 1 µm, 2 µm and 4 µm in ∼ 10 s residence time in the ice super-saturated section of SPIN are calculated according to Rogers and Yau (1989)…*" Can the authors please include the ice crystal growth equation in the manuscript? This will help readers understand the important point they are raising about the kinetic-limitations of the new instrument setup.

**Homogeneous freezing of freezing of Ammonium sulfate solution**

10. Should "*Ammonium*" be capitalized in this section title?

11. In this section title you say "*ammonium sulfate solution,*" but you introduced dried particles into the SPIN. The $(NH_4)_2SO_4)$ particles obviously deliquesced; please briefly report the deliquescence RH in the text, or show it in Figure 4.

12. Page 6 Lines 13-15: "*Possible reasons for the discrepancy between the measurements in this study and the Koop-parametrisation are time dependent effects, i.e. aqueous aerosol do not reach equilibrium before freezing in SPIN.*" See my Major Comment (#1) above. Could the apparent early onset of homogeneous freezing be due to heterogeneities in wall temperature, leading some sections of the aerosol lamina to experience homogeneous freezing conditions while the mean conditions are below homogeneous freezing? Tt would help to report the standard deviation of lamina supersaturation here or show it in Figure 4.

**Discussion**
Page 8 Lines 3-5: "*While the data are generally consistent, the measurements of this study show systematic discrepancies in T-RH dependance both from the predicted heterogeneous ice nucleation of CNT (Fletcher, 1962) and from the widely used parametrisation of homogeneous ice nucleation of solution droplets by Koop et al. (2000). Apart from differences in the method of particle generation, size segregation and detection, it is unclear why partly systematically deviating T-RH dependencies were observed in similar experiments.*" Again, see my comments above about whether uncertainty/variability in the average lamina RH could be responsible for discrepancies in the onset of homogeneous freezing.

**Figure 1**
Caption: The citation here is Murphy and "Koop," not "*Kopp*." Koop's name is also misspelled in the references (Page 11 Line 15).

**Figures 4-5**
The coverage of T and RH space for these experiments is impressive!

**References**
David, R. O., Marcolli, C., Fahrni, J., Qiu, Y., Perez Sirkin, Y. A., Molinero, V., Mahrt, F., Brühwiler, D., Lohmann, U. and Kanji, Z. A.: Pore condensation and freezing is responsible for ice formation below water saturation for porous particles, Proc. Natl. Acad. Sci., 116(17), 8184–8189, doi:10.1073/pnas.1813647116, 2019.

DeMott, P. J., Prenni, A. J., McMeeking, G. R., Sullivan, R. C., Petters, M. D., Tobo, Y., Niemand, M., Möhler, O., Snider, J. R., Wang, Z. and Kreidenweis, S. M.: Integrating laboratory and field data to quantify the immersion freezing ice nucleation activity of mineral dust particles, , 15(1), 393–409, doi:10.5194/acp-15-393-2015, 2015.

Garimella, S., Rothenberg, D. A. D. A., Wolf, M. J. M. J., David, R. O. R. O., Kanji, Z. A. Z. A., Wang, C., Rösch, M. and Cziczo, D. J. D. J.: Uncertainty in counting ice nucleating particles with continuous flow diffusion chambers, Atmos. Chem. Phys., 17(17), 10855–10864, doi:10.5194/acp-17-10855-2017, 2017.

Koop, T., Luo, B., Tsias, A. and Peter, T.: Water activity as the determinant for homogeneous ice nucleation in aqueoussolutions, Nature, 406(6796), 611–614, doi:10.1038/35020537, 2000.

Kulkarni, G. and Kok, G.: Mobile Ice Nucleus Spectrometer, Pacific Northwest Natl. Lab. Richland, WA, 2012.

Marcolli, C.: Deposition nucleation viewed as homogeneous or immersion freezing in pores and cavities, Atmos. Chem. Phys., 14(4), 2071–2104, doi:10.5194/acp-14-2071-2014, 2014.

Nichman, L., Wolf, M., Davidovits, P., Onasch, T. B. T. B., Zhang, Y., Worsnop, D. R. D. R., Bhandari, J., Mazzoleni, C. and Cziczo, D. J. D. J.: Laboratory study of the heterogeneous ice nucleation on black-carbon-containing aerosol, Atmos. Chem. Phys., 19(19), 12175–12194, doi:10.5194/acp-19-12175-2019, 2019.

Wolf, M. J., Coe, A., Dove, L. A., Zawadowicz, M. A., Dooley, K., Biller, S. J., Zhang, Y., Chisholm, S. W. and Cziczo, D. J.: Investigating the Heterogeneous Ice Nucleation of Sea Spray Aerosols Using Prochlorococcus as a Model Source of Marine Organic Matter, Environ. Sci. Technol., 53(3), doi:10.1021/acs.est.8b05150, 2019.

---

## Referee Comment (RC2) · Anonymous Referee #1 · 4 Sep 2020

**General comments:**

This manuscript describes the modification of the commercial instrument SPIN (SPectrometer for Ice Nuclei), which is a device for measuring ice nucleation activity of laboratory-prepared as well as natural ice-nucleating particles. In particular, by modification of the compressor system for cooling the measurement chamber of the instrument, the authors extended the temperature range of the SPIN device. The functionality in the entire (now extended) range of humidity and temperature is studied using model laboratory aerosols of ammonium sulfate and silver iodide. Overall, this is a valuable approach, although at the low temperature there seems to be an offset from the theoretical homogeneous ice nucleation curve. The authors speculate that this may be due to the fact that the aerosol particles do not reach equilibrium before freezing, see comment (6) below. I find this somewhat unsatisfactory and suggest that the authors spend more thought on this (and maybe, if possible, supply some additional sensitiv-

ity measurements). Moreover, because the paper's goal is allowing other scientists to make the same modifications to their SPIN instrument, I suggest a more detailed description/listing of the individual steps in order to become unambiguous.

Formally, the paper text, length, and figures are appropriate. However, I have a request regarding wording: apparently, the authors mix up vapor pressure and partial pressure and I request a correct and consistent usage of these terms, see comments (1) and (3) below.

In summary, the manuscript provides a useful approach and technical modification of an existing instruments, which I consider to be publishable in Atmospheric Measurement Techniques after the comments below have been considered in a revised version.

**Scientific comments:**

(1) Page 2, Line 17; and caption to Fig.1: "Under steady state conditions a linear temperature and vapour pressure gradient establishes between the plates." I think the term vapor pressure is not used correctly and consistently. Vapor pressure is a property of a liquid or solid, and partial pressure is a property of a gas (mixture). The term vapor pressure is used here with its meaning of partial pressure. The IUPAC definition is: "For a mixture of gases the contribution by each constituent is called the partial pressure." In the caption of Fig.1 the term saturation pressure is used, which actually is the (saturation) vapor pressure.

(2) P.4, Figure 2: I would prefer that the figure directly indicated the modifications in the setup, either by colors or by shading etc. As it is now, the modifications are not evident to me. Given that these modifications are the essential novel part of this study, I also strongly recommend a more-detailed point-by-point listing of all modifications, so that any other SPIN user can follow and repeat it with their setup immediately. The latter may be provided in an appendix or supplement.

(3) P.4, L.6/7: "decreasing absolute vapour pressure". What is an ABSOLUTE vapor pressure? I guess you mean total partial pressure, do you? See comment (1) above.

(4) P.5, Figure 3b and abstract and P.8, L.11: According to Fig.3b, the total experimental range to measure ice nucleation (indicated by the hatched area) is extended by 14 K at maximum when compared to the original SPIN range, but certainly less than the 20 K given in the abstract as well as in the conclusion (P.8, L.11). Please correct accordingly.

(5) P.6, L.10: "and 1% AF curve derived using the parametrization of Koop et al. (2000)" How was the Koop-line calculated for 1% activation? The line will depend upon the time interval for which the aerosol particles are exposed to the T and RH conditions. What time interval was used for the calculations?

(6) P.6, L.15: "aqueous aerosol do not reach equilibrium before freezing in SPIN" This is indeed a possibility, and maybe at the lowest temperatures diffusional limitations within the liquid droplets may also become relevant. However, this non-equilibrium state before freezing hampers the accuracy and applicability of SPIN, in particular in the extended low temperature regime. I would hope for more investigations as this is the main additional range of the SPIN instrument introduced in this work.

**Minor and technical comments:**

(7) P.1, L.15: "At intermediate temperatures (236K < T < 273K) heterogeneous ice nucleation above water saturation..." I believe this should be AT OR BELOW water saturation, rather than ABOVE it. Or do you mean to "... above ICE saturation"?

(8) P.1, L.24: Here and at many other places in the text "Often, the dependency of ice nucleation on T, RH by a specific mechanism" I do not like the notation T, RH within a sentence. I guess you mean "... on T and RH by a ..."? Please reword and use consistently throughout text.

(9) P.2, L.29-30: I assume these are the boiling points at standard or ambient pressure,

correct? Please refine wording.

(10) P.3, Figure caption 1: Reference Murphy and Kopp (2005) is misspelled.

(11) P.4, L.7: "4 $\mu$m in ∼10 s residence time" Replace "in" by "within".

(12) P.6, L.3: "are reports of (NH4)2SO4 forming ice heterogeneously at cirrus temperatures" This is only correct for non-deliquesced (i.e. effloresced) particles. I suggest to add the deliquescence line of (NH4)2SO4 to Fig 4b.

(13) P.8, L.11: "We describe a mechanically easy modification" Maybe "simple" is better than "easy"?

(14) P.8, L.18: Replace "AgI" by "AgI particles"

(15) P.8, L.23: There is no link provided to the repository.

---

## Author Comment (AC1) · 28 Oct 2020

**Response to the Comment of Reviewer 1**

We would like to thank Reviewer 1 for their comments and helpful suggestions. We reply to the individual points below.

**General Comments**

*This manuscript describes the modification of the commercial instrument SPIN (SPectrometer for Ice Nuclei), which is a device for measuring ice nucleation activity of laboratory-prepared as well as natural ice-nucleating particles. In particular, by modification of the compressor system for cooling the measurement chamber of the instrument, the authors extended the temperature range of the SPIN device. The functionality in the entire (now extended) range of humidity and temperature is studied using model laboratory aerosols of ammonium sulfate and silver iodide. Overall, this is a valuable approach, although at the low temperature there seems to be an offset from the theoretical homogeneous ice nucleation curve. The authors speculate that this may be due to the fact that the aerosol particles do not reach equilibrium before freezing, see comment (6) below. I find this somewhat unsatisfactory and suggest that the authors spend more thought on this (and maybe, if possible, supply some additional sensitivity measurements). Moreover, because the paper's goal is allowing other scientists to make the same modifications to their SPIN instrument, I suggest a more detailed description/listing of the individual steps in order to become unambiguous.*
*Formally, the paper text, length, and figures are appropriate. However, I have a request regarding wording: apparently, the authors mix up vapor pressure and partial pressure and I request a correct and consistent usage of these terms, see comments (1) and (3) below.*
*In summary, the manuscript provides a useful approach and technical modification of an existing instruments, which I consider to be publishable in Atmospheric Measurement Techniques after the comments below have been considered in a revised version.*

**Scientific Comments**

(1) *Page 2, Line 17; and caption to Fig.1: "Under steady state conditions a linear temperature and vapour pressure gradient establishes between the plates." I think the term vapor pressure is not used correctly and consistently. Vapor pressure is a property of a liquid or solid, and partial pressure is a property of a gas (mixture). The term vapor pressure is used here with its meaning of partial pressure. The IUPAC definition is: "For a mixture of gases the contribution by each constituent is called the partial pressure." In the caption of Fig.1 the term saturation pressure is used, which actually is the (saturation) vapor pressure.*

Following the reviewers recommendation, we replaced "vapour pressure" with "water vapour partial pressure" throughout the text when referring to the gas phase.

(2) *P.4, Figure 2: I would prefer that the figure directly indicated the modifications in the setup, either by colors or by shading etc. As it is now, the modifications are not evident to me. Given that these modifications are the essential novel part of this study, I also strongly recommend a more-detailed point-by-point listing of all modifications, so that any other SPIN user can follow and repeat it with their setup immediately. The latter may be provided in an appendix or supplement.*

To make the needed changes more evident, we added references to Fig. 2 and list the steps in the text as follows:
To reach lower temperatures, the SPIN cooling system has been modified by reconnecting the cold wall, cascade compressor system to deliver R116 refrigerant to both wall plates. The configuration of the modified setup is shown in Fig. 2. In practice, three modifications to the cooling system are needed:

1. At the upper part of the chamber, a junction is added to the high pressure liquid R116 line (lower stage, yellow line in Fig. 2) to connect it to the warm wall plate in parallel to the cold wall.
2. The refrigerant outlet lines of cold and warm wall, where the refrigerant exits the wall plates in the form of low pressure gas (lower stage, blue line in Fig. 2) are joint together to return to the cold 2 compressor.

3. Accounting for the increased volume of R116 needed to cool both wall plates, an additional expansion volume (14ℓ steel tank) is added to the return line (lower stage, blue line in Fig. 2), to give room to the gaseous refrigerant and prevent overpressure when the system is not running.

(3) *P.4, L.6/7: "decreasing absolute vapour pressure". What is an ABSOLUTE vapor pressure? I guess you mean total partial pressure, do you? See comment (1) above.*

As can be seen in Fig. 1, second row, the water vapour partial pressure at a certain relative humidity decreases with decreasing temperature, causing slower ice crystal growth. We changed the sentence accordingly.

(4) *P.5, Figure 3b and abstract and P.8, L.11: According to Fig.3b, the total experimental range to measure ice nucleation (indicated by the hatched area) is extended by 14 K at maximum when compared to the original SPIN range, but certainly less than the 20 K given in the abstract as well as in the conclusion (P.8, L.11). Please correct accordingly.*

The stated "more than 20K increase in measurement range" is based on experience and depends on ambient factors as well as operator's decisions. We consider it a good estimate, to give a SPIN owners an idea of what to expect if they decide to do the modification. Shown as hatched area in Fig. 3(a) and Fig. 1 below, in theory (neglecting heat transfer from the ambient), the temperature range achievable at the lamina position increases between 30K at $RH_{ice} = 100\%$ and 14K at $RH_{ice}$=190%. There are two practical factors to consider: First, operating SPIN in a laboratory or field station at around 293K ambient temperature limits the lowest achievable wall plate temperatures to 5-10K above the boiling point of the refrigerant due to imperfect insulation of the wall plates. Exemplary measurements conditions are shown in Fig. 1 below. Decreasing the cooling rate and ambient temperature can reduce this offset, but are often impractical. Secondly, not all refrigerant changes phase when the chamber is operated at a temperatures too close to the boiling point of the refrigerant. This causes incompressible liquid refrigerant to flow back to the compressor, causing what is known as "liquid slugging" that destroys the compressor. The "safe" temperature range of operating the SPIN chamber is therefore offset, both in the original and the modified configuration, by 5-10K from the maximum. Taking the two factors into account, shifts the calculated ranges (hatched in Fig. 1, below), 5-10K towards higher temperatures and larger ice crystals.

[Figure]

**Figure 1.** Comparison of the typical range of experimental conditions probable with the original SPIN (red points) to conditions after the modification (green points, lower 20 K of data shown in Fig. 4(a)).

The "limiting conditions" from ice crystal grow in Fig. 3(b) assume spherical ice growth (equations are now added in appendix A) and are therefore a lower limit for ice growth within the residence time in SPIN. Time dependent ice nucleation on the other hand can introduce a spread in growth time of the forming ice crystals and thereby their size distribution. It is an operator's decision which sizes are counted as ice crystals. From practical experience, the modification increased the measurement range by $> 20K$ at low $RH_{ice}$ and $> 15K$ at high $RH_{ice}$. The temperature range to conduct RH-scans at constant T is increased by $> 20K$. The statement of a gain in temperature range of over 20K is therefore kept as is in the conclusion. In the abstract we now state the lowest temperature ($208\,K$) at which we performed experiments, instead.

(5)  *P.6, L.10: "and 1% AF curve derived using the parametrization of Koop et al. (2000)" How was the Koop-line calculated for 1% activation? The line will depend upon the time interval for which the aerosol particles are exposed to the T and RH conditions. What time interval was used for the calculations?*

The equations to calculate the 1% AF using the parametrisation of Koop et al. (2000) have been added in appendix B. We use the total residence time of exposure to the respective conditions (residence times slightly change with operating conditions and are calculated individually at each point in the RH-T space) to obtain the droplet size. The line of homogeneous freezing in Fig. 4(b) gives an estimation for the lowest $RH_{ice}$ at each temperature where homogeneous freezing can cause 1% of droplets to freeze.

(6)  *P.6, L.15: "aqueous aerosol do not reach equilibrium before freezing in SPIN" This is indeed a possibility, and maybe at the lowest temperatures diffusional limitations within the liquid droplets may also become relevant. However, this non-equilibrium state before freezing hampers the accuracy and applicability of SPIN, in particular in the extended low temperature regime. I would hope for more investigations as this is the main additional range of the SPIN instrument introduced in this work.*

At this point we would like to consider the discrepancy an interesting observation and only speculate on the reason. From the scatter of literature data included in Fig. 4(b) it can be seen that deciding whether the experimental data or parametrization (partly based on this data) are more robust is difficult. If droplets were not in equilibrium, they would be smaller and more concentrated, resulting in the 1% AF line to shift towards higher RH. This would explain the observed offset at the lower temperatures. Alternatively, Cziczo and Abbatt (1999) argued against substantial excursion from equilibrium even at short timescales and low temperatures, leaving the interpretation that the Koop et al. (2000) parametrization has a small, eventually solute dependent (Swanson, 2009) offset. We disagree that the applicability of SPIN to investigate homogeneous freezing is affected. The SPIN setup can provide measurements to investigate and refine the current understanding, be it diffusion of water into particles or ice nucleation.

**Minor and Technical Comments**

(7)  *P.1, L.15: "At intermediate temperatures (236K < T < 273K) heterogeneous ice nucleation above water saturation. . ." I believe this should be AT OR BELOW water saturation, rather than ABOVE it. Or do you mean to ". . . above ICE saturation"?*

We changed the wording to "... close to water saturation..."

(8)  *P.1, L.24: Here and at many other places in the text "Often, the dependency of ice nucleation on T, RH by a specific mechanism" I do not like the notation T, RH within a sentence. I guess you mean ". . . on T and RH by a . . ."? Please reword and use consistently throughout text.*

"T, RH" was replaced by "T and RH" when used within a sentence.

(9)  *P.2, L.29-30: I assume these are the boiling points at standard or ambient pressure, correct? Please refine wording.*

We added that boiling points are given at atmospheric pressure ($1\,atm$).

(10)  *P.3, Figure caption 1: Reference Murphy and Kopp (2005) is misspelled.*

Corrected.

(11)  *P.4, L.7: "4 m in  10 s residence time" Replace "in" by "within".*

Corrected.

(12) *P.6, L.3: "are reports of $(NH_4)_2SO_4$ forming ice heterogeneously at cirrus temperatures" This is only correct for non-deliquesced (i.e. effloresced) particles. I suggest to add the deliquescence line of $(NH_4)_2SO_4$ to Fig 4b.*

We added the temperature dependent DRH line to Fig. 4(b).

(13)  *P.8, L.11: "We describe a mechanically easy modification" Maybe "simple" is better than "easy"?*

We replaced "easy" with "simple".

(14)  *P.8, L.18: Replace "AgI" by "AgI particles"*

Corrected.

(15) *P.8, L.23: There is no link provided to the repository.*

Unfortunately our data repository is not operational yet. The data are available upon request from the author.

**References**

Cziczo, D. J. and Abbatt, J. P. D.: Deliquescence, efflorescence, and supercooling of ammonium sulfate aerosols at low temperature: Implications for cirrus cloud formation and aerosol phase in the atmosphere, J. Geophys. Res., pp. 13 781 – 13 790, doi:10.1029/1999JD900112, 1999.

Koop, T., Luo, B. P., Tsias, A., and Peter, T.: Water activity as the determinant for homogeneous ice nucleation in aqueous solutions, Nature, 406, 611–614, 2000.

Swanson, B. D.: How Well Does Water Activity Determine Homogeneous Ice Nucleation Temperature in Aqueous Sulfuric Acid and Ammonium Sulfate Droplets?, J. Atmos. Sci., pp. 741 – 754, doi:https://doi.org/10.1175/2008JAS2542.1, 2009.

---

## Author Comment (AC2) · 28 Oct 2020

**Response to the Comment of Reviewer 2**

We would like to thank Reviewer 2 for their comments and helpful suggestions. We reply to the individual points below.

**General Comments**

*The paper summarizes a design modification that will be of significant interest to other SPIN users and most likely other continuous flow diffusion chamber (CFDC) designs as well. Their technique has several demonstrable benefits. The authors note that the SPIN can now operate within the full set of conditions relevant to cirrus formation in the upper troposphere. One additional benefit not explicitly highlighted by the authors is the reduced number of compressors. Previous attempts to operate CFDCs at cold temperatures have resulted in the damage of compressors, and reducing the number of compressors used in the design will likely reduce instrument down-time due to repairs.*
*This paper is therefore well-suited for publication in AMT. Below, I provide a few questions and comments to strengthen the paper and clarify the interpretation of the SPIN results. The authors will note that most of these are minor points. I therefore recommend the paper be accepted for publications after the authors have adequately addressed or responded to them.*

**Major Comments**

(1) *My only major comment concerns the interpretation of the apparent early-onset of homogeneous freezing of (NH4)2SO4). The authors note that the (NH4)2SO4) solution droplets appear to nucleate at a lamina RH lower than that expected for homogeneous freezing (Koop et al., 2000). The authors claim a discrepancy between their data and the Koop parameterization. However, I would venture a guess that their data actually presents no discrepancy if the uncertainty in lamina RH is taken into account.*

   a. *I believe Fig. 4 reports the average lamina conditions. However, at high RH and cold temperatures, CFDCs (and I would guess SPIN) generally show a few % uncertainty in the lamina RH. This uncertainty is caused by variability in the wall temperatures. Colder regions of the wall – for example, where refrigerant is injected – can cause certain areas of the lamina to experience a higher RH than the average. See e.g. Kulkarni and Kok, 2012, for a simple method and pre-written code to calculate the variability in lamina RH for a CFDC with SPIN's geometry.*

   The experimental variability in RH and T has been added to Fig. 4(b) and 5(b). The indicated range of variability represents the maximal deviation from the average conditions along the sample lamina, obtained from RH and T profiles calculated between 15 pairs of opposite temperature measurements along the cold and warm wall.

   b. *If this is lamina range is taken into account, does the onset of (NH4)2SO4) nucleation more closely align with the Koop homogeneous freezing parameterization?*

   Taking the RH and T variability into consideration does not change the observation that the measured slope of homogeneous freezing conditions is steeper than the Koop-line.

**Minor Comments**
**Abstract**

(2) *Page 1 Lines 4-5: "The modification extends the measurement range of SPIN by more than 20 K to the temperature regime relevant for ice formation in cirrus clouds." Can the authors specify in the abstract the lower temperature range now achievable with their design modification?*

   We now specify the lowest temperature (208 K) at which measurements were performed. Measurements at lower temperatures are possible, but generate detection issues of distinguishing the sample aerosol from ice crystals.

**Introduction**

(3) *Page 1 Lines 13-15: "Tropospheric ice nucleation at low temperatures (T < 236 K), typical for cirrus clouds, proceeds at water sub-saturated conditions by homogeneous nucleation of aqueous aerosol or heterogeneous nucleation from the vapour phase." The authors should briefly mention the possibility that heterogeneous nucleation below water supersaturation could be due to the pore-condensation freezing mechanism. E.g. (David et al., 2019; Marcolli, 2014)*

Pore condensation freezing is already mentioned explicitly further down in this paragraph (line 19) and a reference to the recent overview by Marcolli, 2020 is given in line 25.

**Operating Principles**

(4) *Page 2 Lines 17-18: "For an explicit derivation of the linear temperature and vapour pressure field in a CFDC we refer to Rogers (1988); Luond (2009)." I recommend also citing here Kulkarni and Kok, 2012 – it specifically discusses calcualtion of lamina conditions for the parallel plate (SPIN) design.*

We thank the reviewer for pointing to this useful publication. We added: "A ready to use code to determine the position of the sample lamina can be found in Kulkarni and Kok (2012)."

(5) *Page 2 Line 21: "...a lamellar sample, which is confined by a sheath flow to a narrow position between the ice covered wall plates..." The authors should briefly note here recent work that demonstrates aerosol samples are \*not\* constrained by sheath flows but rather spread outside the lamina. See e.g. DeMott et al., 2015; Garimella et al., 2017. This fact should not much change the author's results or interpretation, but it is important for the field of CFDC users to start to acknowledge.*

We agree that the fraction of particle leaving the lamina does not considerably change the results and interpretation. We added: "For a discussion on sampling bias due to particle displacement outside the lamina we refer to Garimella et al. (2017); Korhonen et al. (2020)."

**Modified Cooling System**

(6) *Page 3 Line 31: "To reach lower temperatures, the SPIN cooling system has been modified by reconnecting the cold wall, cascade compressor system to deliver R116 refrigerant to both wall plates." The authors may know that R116 is a HFC whose use is being phased out in the European Union as per the Kigali Amendment to the Montreal Protocol. At this time, do the authors have knowledge of any non-HFC refrigerant (e.g. hydrofluoroolefin, CO2) that might be an acceptable substitute in the future? The authors might note that if HFC refrigerants are banned, a new overall to the SPIN or other CFDC instruments' refrigeration loops may be needed anyways.*

The use of HFC as refrigerant is a concerning issue, even in small quantities for scientific purposes. R116 as low-temperature refrigerant has already been phased out in the EU. We have been searching for non-HFC refrigerants that have a boiling point near the one of R116, but at present there are few suitable substitutes available. The referee is right that, e.g. $CO_2$ (boiling point 195 K) might suit for retrofitting the SPIN, but it will need to be tested.

(7) *Page 3 Lines 4-6: "A consequence of using only one instead of two compressors to deliver the refrigerant for both walls, is a reduction in the achievable cooling rate from approximately 2 K min1 to 1 K min1 above 233 K and decreasing to < 0.5 Kmin1 towards the lowest temperatures." Please comment on the circumstances when both walls need to be cooled concurrently (i.e. during cooling down to start experiment).*

The reduced cooling rate refers to the situation when both walls are cooled simultaneously. We added: "Simultaneous cooling of both walls is needed during cooling of the chamber to start an experiment or measurements in which temperature is changed at a constant humidity (T-scan)."

(8) *Page 3 Lines 11-13: "The range of the original SPIN setup is calculated with the cold wall varying from 273.15 K-194.95 K and the warm wall between 273.15 K-226.65 K (boiling point of R404A)." The authors should note how this range compares to the coldest temperatures previously achieved for SPIN experiments. The coldest I can find published are -58 °C (Wolf et al., 2019) and -56 °C (Nichman et al., 2019).*

The only possibility to achieve these low temperatures in the original SPIN, is by applying an asymmetrical sheath flow, pushing the sample lamina towards the cold wall. As can be seen in Fig. 1 in the manuscript, this leads to a not well constrained RH within the lamina. The two mentioned articles did not reveal the method they used to achieve low temperatures, therefore we prefer not to speculate.

(9) *Page 4 Lines 7-8: "Conditions for ice crystals to grow to a diameter of 0.5 m, 1 m, 2 m and 4 m in 10 s residence time in the ice super-saturated section of SPIN are calculated according to Rogers and Yau (1989)..." Can the authors please include the ice crystal growth equation in the manuscript? This will help readers understand the important point they are raising about the kinetic-limitations of the new instrument setup.*

The equation to calculate the ice crystal size is now included in appendix A.

**Homogeneous freezing of freezing of Ammonium sulfate solution**

(10) *Should "Ammonium" be capitalized in this section title?*

Changed to "ammonium".

(11) *In this section title you say "ammonium sulfate solution," but you introduced dried particles into the SPIN. The (NH4)2SO4) particles obviously deliquesced; please briefly report the deliquescence RH in the text, or show it in Figure 4.*

The DRH-line has been added to Fig. 4(b).

(12) *Page 6 Lines 13-15: "Possible reasons for the discrepancy between the measurements in this study and the Koop-parametrisation are time dependent effects, i.e. aqueous aerosol do not reach equilibrium before freezing in SPIN." See my Major Comment (1) above. Could the apparent early onset of homogeneous freezing be due to heterogeneities in wall temperature, leading some sections of the aerosol lamina to experience homogeneous freezing conditions while the mean conditions are below homogeneous freezing? Tt would help to report the standard deviation of lamina supersaturation here or show it in Figure 4.*

The variability in experimental conditions is now shown in Fig. 4(b) and Fig. 5(b). The standard deviation when repeating the experiment several times is much smaller.

**Discussion**
*Page 8 Lines 3-5: "While the data are generally consistent, the measurements of this study show systematic discrepancies in T-RH dependance both from the predicted heterogeneous ice nucleation of CNT (Fletcher, 1962) and from the widely used parametrisation of homogeneous ice nucleation of solution droplets by Koop et al. (2000). Apart from differences in the method of particle generation, size segregation and detection, it is unclear why partly systematically deviating T-RH dependencies were observed in similar experiments." Again, see my comments above about whether uncertainty/variability in the average lamina RH could be responsible for discrepancies in the onset of homogeneous freezing.*

We added the variability in the lamina RH and T to Fig. 4(b). While the Koop-line is within the variability of conditions during an experiment, the slope of the conditions clearly deviate from theory. Additional investigations are needed to elucidate this observation.

**Figure 1**
*Caption: The citation here is Murphy and "Koop," not "Kopp." Koop's name is also misspelled in the references (Page 11 Line 15).*

Corrected.

**Figure 4-5**
*The coverage of T and RH space for these experiments is impressive!*

**References**

Garimella, S., Rothenberg, D. A., Wolf, M. J., David, R. O., Kanji, Z. A., Wang, C., Rösch, M., and Cziczo, D. J.: Uncertainty in counting ice nucleating particles with continuous flow diffusion chambers, Atmos. Chem. Phys., pp. 10 855 – 10 864, doi:https://doi.org/10.5194/acp-17-10855-2017, 2017.

Korhonen, K., Kristensen, T. B., Falk, J., Lindgren, R., Andersen, C., Carvalho, R. L., Malmborg, V., Eriksson, A., Boman, C., Pagels, J., Svenningsson, B., Komppula, M., Lehtinen, K. E. J., and Virtanen, A.: Ice-nucleating ability of particulate emissions from solid-biomass-fired cookstoves: an experimental study, Atmos. Chem. Phys., pp. 4951 – 4968, doi:https://doi.org/10.5194/acp-20-4951-2020, 2020.

Kulkarni, G. and Kok, G.: Mobile Ice Nucleus Spectrometer, techreport PNNL-21384, Pacific Northwest Natl. Lab. Richland, 2012.